# Diet-Induced Gut Barrier Dysfunction Is Exacerbated in Mice Lacking Cannabinoid 1 Receptors in the Intestinal Epithelium

**DOI:** 10.3390/ijms231810549

**Published:** 2022-09-11

**Authors:** Mark B. Wiley, Nicholas V. DiPatrizio

**Affiliations:** Division of Biomedical Sciences, School of Medicine, University of California-Riverside, Riverside, CA 92521, USA

**Keywords:** cannabinoid receptor-1, diet-induced obesity, gut barrier

## Abstract

The gut barrier provides protection from pathogens and its function is compromised in diet-induced obesity (DIO). The endocannabinoid system in the gut is dysregulated in DIO and participates in gut barrier function; however, whether its activity is protective or detrimental for gut barrier integrity is unclear. We used mice conditionally deficient in cannabinoid receptor subtype-1 (CB_1_R) in the intestinal epithelium (intCB1−/−) to test the hypothesis that CB_1_Rs in intestinal epithelial cells provide protection from diet-induced gut barrier dysfunction. Control and intCB1−/− mice were placed for eight weeks on a high-fat/sucrose Western-style diet (WD) or a low-fat/no-sucrose diet. Endocannabinoid levels and activity of their metabolic enzymes were measured in the large-intestinal epithelium (LI). Paracellular permeability was tested in vivo, and expression of genes for gut barrier components and inflammatory markers were analyzed. Mice fed WD had (i) reduced levels of endocannabinoids in the LI due to lower activity of their biosynthetic enzymes, and (ii) increased permeability that was exacerbated in intCB1−/− mice. Moreover, intCB1−/− mice fed WD had decreased expression of genes for tight junction proteins and increased expression of inflammatory markers in LI. These results suggest that CB_1_Rs in the intestinal epithelium serve a protective role in gut barrier function in DIO.

## 1. Introduction

The gut barrier is essential for providing protection from pathogens and consists of three primary layers: (i) a mucous barrier containing IgA antibodies and mucins, (ii) a single cell layer of epithelial cells lining the small and large intestine, and (iii) transmembrane tight junction proteins that provide an additional layer of protection between epithelial cells [1]. Disruption of this barrier is associated with inflammation and increased susceptibility to infection [2]. In severe cases of barrier dysfunction (e.g., inflammatory bowel diseases), apoptosis of epithelial cells lining the intestinal tract occurs, which can lead to an enhanced inflammatory response and significant damage to intestinal tissue [3]. In more subtle cases of gut barrier dysfunction—such as in obesity—expression of transmembrane tight junction proteins are down-regulated, which can lead to increased passage of pathogens and small molecules between epithelial cells and into circulation [4]. For example, consumption of a high-fat and high-sucrose Western-style diet is associated with a mild increase in gut barrier permeability in mice, which suggests that diet may impact integrity of the gut barrier [5,6].

A variety of signaling pathways in the intestinal epithelium are implicated in its function and dysfunction in health and disease, including the endocannabinoid system [7]. For example, activity of this endogenous lipid signaling system in the upper small-intestinal epithelium is increased in a mouse model of Western diet-induced obesity and drives overeating by dampening nutrient-induced gut-brain satiation signaling [7,8]. Moreover, the endocannabinoid system in the large intestine, including cannabinoid receptor subtype 1 (CB_1_) and 2 (CB_2_), may participate in gut barrier function and inflammation [9,10,11,12,13,14,15,16,17,18,19,20,21,22,23,24,25]. It remains unclear, however, if activating the endocannabinoid system is either protective or detrimental to gut barrier function given conflicting results that are dependent upon both the experimental model and the methods associated with application of cannabinoid compounds (e.g., basolateral vs. apical) [9,11,12,17,18,19]. Nonetheless, these studies suggest complex interactions between the endocannabinoid system, gut permeability, and inflammation, and the need for reliable and reproducible experimental models to identify specific roles for the gastrointestinal endocannabinoid system in health and disease, and the mechanisms involved in these functions.

Several experimental models are used to induce disruption of gut barrier function, including chemicals that cause apoptosis of epithelial cells along the intestinal tract (i.e., dextran sodium sulfate, DSS mouse model of colitis), which stimulates a strong inflammatory response at the site of damage [2]. In contrast to the DSS mouse model of colitis, diet-induced obese mice provide a less severe model for inducing gut barrier dysfunction while largely maintaining tissue integrity. Diet-induced obesity is associated with increased paracellular permeability in the large intestine by mechanisms that include down regulation of tight junction protein expression (e.g., ZO-1, occludin) and increased local inflammation (e.g., IL-6, TNFα, T-cell infiltrate) [4,26]. Accordingly, diet-induced obese mice provide a useful model for identifying discrete changes in gut barrier function, and associated molecular pathways, without compromising the general architecture of the intestinal tract.

We used a mouse model of Western diet-induced obesity (i.e., chronic access to a high-fat and high-sucrose laboratory diet) in combination with transgenic mice that conditionally lack CB_1_Rs in the intestinal epithelium (intCB1−/−) to identify roles for the endocannabinoid system in the intestinal epithelium in diet-induced gut barrier dysfunction. We evaluated the effects of diet and genotype on (i) gut barrier function, and (ii) inflammatory and related immune function in the large intestinal epithelium. These studies advance our understanding of important roles for the endocannabinoid system in gut barrier function in health and disease (see Figure 1 for summary).

## 2. Results

### 2.1. Chronic Consumption of Western Diet Is Associated with Reduced Levels of Endocannabinoids and Activity of Their Biosynthetic Enzymes in the Large-Intestinal Epithelium

Levels of the endocannabinoid, 2-AG, and several other related monoacylglycerols (MAGs) were lower in the large-intestinal epithelium from mice fed WD for eight weeks when compared to mice fed SD (Table 1). Levels of the satiety-related fatty acid ethanolamide, oleoylethanolamide (OEA, [27]), were also lower in mice fed WD. We next aimed to identify if reductions in levels of 2-AG and other MAGs in mice fed WD were due to changes in activity of enzymes responsible for their metabolism. We analyzed separately ex vivo activity of enzymes responsible for the biosynthesis of MAGs (e.g., diacylglycerol lipase, DGL, [28]) and degradation of MAGs (e.g., monoacylglycerol lipase, MGL), in the large-intestinal epithelium. When compared to mice fed SD, those fed WD displayed significant decreases in activity of MAG biosynthetic enzymes (Figure 2A, *p* < 0.05), and no significant changes in activity of MAG degradative enzymes (Figure 2B *p* = 0.85). Together, these results identify a dysregulation of eCB biosynthesis in the large-intestinal epithelium in mice chronically fed WD, which may contribute to disrupted gut barrier function in diet-induced obesity. A summary of statistical analyses and results are included in Appendix A.

### 2.2. CB_1_Rs in the Intestinal Epithelium Control Gut Barrier Permeability in Mice Fed WD

We next aimed to identify roles for CB_1_Rs in the large-intestinal epithelium in controlling gut permeability in mice maintained on WD for eight weeks or control mice maintained on SD. We first analyzed via qPCR expression of mRNA in the large-intestinal epithelium for several common tight junction proteins in intCB_1_+/+ and intCB1−/− mice maintained on SD (baseline recording at 10 days following the final tamoxifen administration). No changes in expression of genes were found between groups for tight junction protein-1 (Tjp-1, *p* = 0.85), occludin (Ocln, *p* = 0.37), and claudin-1 (Cldn-1, *p* = 0.55). Validation of knockdown of genes for CB_1_Rs in the large intestinal epithelium in intCB1−/− mice was confirmed (Cnr1, *p* < 0.05). A summary of statistical analyses and results are included in Appendix A.

We then examined gut paracellular permeability by analyzing levels of FITC-Dextran in serum from separate groups of intCB1+/+ and intCB1−/− mice maintained on SD or WD for up to eight weeks. All mice maintained on WD weighed significantly higher than those maintained on SD beginning at week three following initial access to WD (Figure 3A). Prior to being placed on corresponding diets, gut permeability was tested across genotypes, which revealed no significant differences in levels of FITC-Dextran (Figure 3B, *p* = 0.43). Moreover, after two weeks on SD or WD, no significant changes occurred in levels of FITC-dextran across genotypes or diet (Figure 3C). After eight weeks on corresponding diets, however, paracellular permeability was significantly compromised with concomitant increases in levels of FITC-dextran in serum in intCB1+/+ mice fed WD when compared to intCB1+/+ mice fed SD (Figure 3D), *p* < 0.05). Furthermore, levels of FITC-dextran in serum were also elevated in intCB1−/− mice fed WD for eight weeks when compared to intCB1−/− mice fed SD (*p* < 0.001). Notably, however, this effect was significantly exacerbated with a near doubling of levels in intCB1−/− mice fed WD when compared to intCB1+/+ mice fed WD for eight weeks (*p* < 0.001). A summary of statistical analyses and results are included in Appendix A.

### 2.3. No Changes in General Morphology of the Large Intestine in Mice Lacking CB_1_Rs in the Intestinal Epithelium

All mice fed WD for eight weeks, irrespective of genotype, had (i) significantly increased epididymal fat pad weight when compared to mice maintained on SD (Figure 4A, effect of diet only, *p* < 0.001), and (ii) significant reductions in length of the large intestine (Figure 4B, effect of diet only, *p* < 0.05). In contrast, irrespective of both diet or genotype, no changes were found in weight of the large intestine or weight-to-length ratio (Figure 4C,D, respectively). No discernable qualitative changes in tissue morphology were observed across diet or genotype (Figure 4E–H). A summary of statistical analyses and results are included in Appendix A.

### 2.4. Increased Inflammatory Response in the Large-Intestinal Epithelium in Mice Lacking CB_1_Rs in the Intestinal Epithelium and Fed WD

NanoString Sprint Profiler technology was next used to identify transcriptomic changes in genes associated with gut barrier function, inflammatory processes, and components of the endocannabinoid system in the large-intestinal epithelium from intCB_1_+/+ mice and intCB1−/− mice fed SD or WD for eight weeks. Principal Component Analysis (PCA) revealed generally close clustering of samples from intCB_1_+/+ mice fed SD, intCB1−/− mice fed SD, and intCB_1_+/+ mice fed WD regardless of diet or genotype. Notably, however, unique clustering of intCB1−/− mice fed WD was observed across both genotype and diet indicating a unique gene expression profile within this specific group (Figure 5A,B). A representative heatmap of all 18 differentially expressed genes (DEGs) across diet or genotype indicated an upregulation of inflammatory-associated transcripts and a downregulation in barrier-associated transcripts in intCB1−/− mice fed WD when compared to intCB_1_+/+ mice fed WD (Figure 5C).

We next performed a DAVID analysis to determine the number of DEGs that were implicated in specific pathways while simultaneously determining how strongly correlated the DEGs and pathways were using a false discovery rate threshold of -LOG(FDR) > 2. Several distinct pathways were affected by genotype and diet, including those associated with several common molecular functions (Figure 6A), cellular components (Figure 6B), and biological processes (Figure 6C).

Cldn19 was the only gene associated with tight junction proteins and gut barrier integrity found to be significantly altered in large-intestinal epithelium across genotypes in mice fed SD (Figure 6D). When comparing effects of diet in intCB_1_+/+ mice fed WD versus intCB_1_+/+ control mice fed SD, those fed WD had reduced expression of several genes associated with gut barrier function, including Cldn8, Cldn12, and Cldn19 in the large-intestinal epithelium (Figure 6E).

Several additional DEGs were identified in the large-intestinal epithelium that are associated with inflammation and immune function, including an upregulation in intCB1−/− mice fed WD, when compared to intCB1−/− mice fed SD, of CXCR5, CXCR3, CXCR6 HLA-DQA1, HLA-DRA, ICAM1, CD247, CXCL13, and IL1A, and a downregulation in expression of IFNGR1 and DAGLBETA, the β isoform of the MAG biosynthetic enzyme, monoacylglycerol lipase (Figure 6F). A similar number of DEGs were found when comparing intCB_1_+/+ mice fed WD to intCB1−/− mice fed WD, with an increase in the latter in several associated with inflammation and immune function, including ICAM1, CXCR6, IL1A, HLA-DRA, CD247, HLA-DQA1, and a decrease in IFNGR1 (Figure 6G). In addition, when compared to intCB_1_+/+ mice fed WD, expression of genes for tight junction proteins were decreased in intCB1−/− mice fed WD, including TJP2, TJP3, CLDN7, and increases in expression of MAGL, a primary MAG degradative enzyme (Figure 6H). A summary of statistical analyses and results are included in Appendix A.

## 3. Discussion

These data provide evidence that chronic access to WD is associated with (i) dysregulation of the endocannabinoid system in the large intestinal epithelium via reduced activity of local biosynthetic enzymes for MAGs and ensuing reductions in levels of 2-AG, (ii) detrimental changes in gut barrier function and inflammation, and (iii) an exacerbated increase in gut paracellular permeability in mice lacking CB_1_Rs in cells in the intestinal epithelium due to mechanisms that may include reductions in gut barrier integrity and increases in local inflammation. Together, these experiments suggest that CB_1_Rs in intestinal epithelial cells control gut barrier function and associated inflammation, and their activity exerts a protective influence during WD-induced obesity in mice.

### 3.1. In Vivo Considerations

The endocannabinoid system participates in gut barrier function and inflammation; however, it is unclear if its activity in the intestinal epithelium serves a protective or detrimental role in these processes. Prior in vivo rodent studies suggest protective effects for both activation and inhibition of the endocannabinoid system that are dependent on the experimental model and protocol. For example, inflammation and intestinal damage in rodent models of colitis (i.e., oil of mustard-induced colitis [13] or trinitrobenzene sulfonic acid (TNBS)-induced colitis [14,15,24]) were reduced after pharmacological activation of CB_1_Rs and CB_2_Rs, or exacerbated in mice globally lacking these receptors [23]. Furthermore, treatment of diet-induced obese mice with the bacteria, *Akkermansia muciniphila*, increased levels of 2-AG and other monoacylglycerols in the distal small intestine and ameliorated gut barrier dysfunction, which also suggests that activation of the endocannabinoid system is protective in a mouse model of obesity [17]. Similarly, administration of inhibitors of endocannabinoid degradation were effective at restoring integrity of intestinal gut barrier function in TNBS-induced colitis [12]. In contrast to these studies that highlight beneficial effects for activating the endocannabinoid system, other studies show that pharmacological activation of cannabinoid receptors in lean mice led to increases in plasma levels of lipopolysaccharide (LPS), which is an endotoxin released from Gram-negative bacteria, and its high presence in circulation suggests compromised gut barrier function [18]. Additionally, pharmacological inhibition of CB_1_Rs in genetically-obese (*ob*/*ob*) [18] and diet-induced obese mice [19] reduced levels of plasma LPS and inflammatory responses, which suggests that inhibiting endogenous activity at CB_1_Rs can be protective against obesity-induced gut barrier dysfunction and associated inflammation.

### 3.2. In Vitro Considerations

Several in vitro studies also highlight differential effects on intestinal permeability for phytocannabinioids versus endocannabinoids and the method of application (i.e., apical versus basolateral). For example, cytokines IFNγ and TNFα increased permeability of Caco-2 cell monolayers measured by transepithelial resistance (TEER, a functional marker of gut barrier integrity), an effect reversed by apical—but not basolateral—application of Δ^9^THC or cannabidiol (CBD) [11]. These results suggest that activating the endocannabinoid system in vitro may serve a protective role in cytokine-induced gut barrier dysfunction. In contrast to phytocannabinoids, endocannabinoids (anandamide or 2-AG) increased permeability when applied to the apical membrane, but had no effect when applied to the basolateral membrane. Moreover, incubation with inhibitors of endocannabinoid degradative enzymes was associated with increased cytokine-induced permeability [9,11]. In contrast, when 2-AG was applied basolaterally to Caco-2 cells alone without cytokines, TEER was increased suggesting an improvement in permeability. Moreover, the effects of phytocannabinoids or endocannabinoids were blocked by co-incubation with a CB_1_R antagonist, but not a CB_2_R antagonist, and a CB_1_R antagonist alone improved permeability when applied to the apical membrane, which highlights roles for CB_1_Rs in these processes. Further implicating CB_1_Rs in the effects of cannabinoids on permeability, inhibitors of endocannabinoid degradation or endocannabinoids alone had no effects on TEER in Caco-2 cells deficient in CB_1_Rs [9].

In contrast to these experiments, both apical or basololateral incubation of Caco-2 cells with Δ^9^THC or CBD improved elevated permeability induced by ethylenedialenetetraacetic acid (EDTA) via a CB_1_R-dependent mechanism [10]. Similar to the cytokine-induced model of increased permeability, endocannabinoids increased permeability when applied to the apical membrane in the EDTA model; however, endocannabinoids were successful in improving recovery of permeability when applied to the basolateral membrane. Both phytocannabinoids and endocannabinoids increased expression of genes for the tight junction protein, ZO-1, while endocannabinoids reduced expression of claudin-1 [10]. Furthermore, incubation of Caco-2 Cells with HU-210, a synthetic cannabinoid agonist, led to decreased expression of genes for ZO-1 and occludin via CB_1_Rs, which suggests that activating CB_1_Rs increases gut permeability [18]. Moreover, Cuddihey and colleagues recently reported that both the cannabinoid receptor agonist, CP55,940, and the neutral CB_1_R antagonist, AM6545, reduced permeability ex vivo in an Ussing Chamber containing jejunum or ileum from mice fed a high fat diet for two weeks, and these effects were absent in intestine from global CB_1_R-null mice [29]. Neither drug altered permeability in mice fed a standard rodent chow, which highlights the impact of diet on these processes.

Together, these studies show that cannabinoids have direct effects on cells in the intestinal epithelium, and the local endocannabinoid system participate in gut barrier function. Similar to in vivo experiments detailed previously, however, these responses differ depending upon the model or cannabinoid compound used, which highlights the need for additional models to examine roles for the intestinal endocannabinoid system in gut function and dysfunction. In addition, these studies were largely unable to discern effects of cannabinoid receptor ligands on receptors expressed in cells in the intestinal epithelium versus those in immune cells or possibly local bacteria [25,30]. For example, mice with genetic deletion of the MAG degradative enzyme, MGL, displayed increases in levels of 2-AG in the large intestine that is accompanied by protection from enteric bacterial infection [30]. This occurred by a mechanism that is suggested to include 2-AG-mediated inhibition of the bacterial receptor, QseC, which in turn, blocks the establishment of virulence and ensuing infection. Furthermore, studies by Acharya and colleagues showed that anandamide regulates immune homeostasis in the gut by promoting presence of immunosuppressive macrophages (i.e., CX3CR1^hi^) [25]. Accordingly, our intCB1−/− mice provides an in vivo model to directly test roles in gut barrier function and inflammation for CB_1_Rs expressed selectively in cells in the intestinal epithelium, which is not possible with traditional pharmacological approaches or whole-body CB_1_R-null mice.

### 3.3. Model of Conditional CB_1_R Deletion and Impact on Gut Barrier Function

To address inconsistencies in the literature discussed above and to elucidate roles for endocannabinoid signaling specifically at CB_1_Rs in intestinal epithelial cells, we placed intCB1−/− and intCB_1_+/+ mice on WD for eight weeks and tested in vivo gut-barrier permeability. Irrespective of genotype, no changes were found in baseline expression of mRNA for the gut barrier proteins tight-junction protein 1 [also known as zona occludins-1; ZO-1)], occludin, or claudin-1. Furthermore, intCB_1_R^−/−^ mice also displayed no changes in FITC-Dextran levels in serum at baseline or after two weeks on WD when compared to mice maintained on a SD, indicating no initial shift in gut barrier function in vivo. After eight weeks on WD, however, both intCB1−/− and intCB_1_+/+ mice displayed significant increases in serum FITC-Dextran when compared to mice fed SD, which, in contrast, had no changes in gut permeability. Notably, impairments in gut permeability were exacerbated in intCB1−/− mice fed WD for eight weeks, which suggests a more severely compromised gut barrier in the absence of CB_1_Rs in intestinal epithelial cells in DIO mice. Despite marked changes in gut barrier permeability, significant changes in tissue morphology that often accompany more severe models of gut barrier dysfunction (i.e., DSS, TNBS; [2,31]) were not observed. Moreover, levels of FITC-dextran in blood remained similar throughout testing between intCB1−/− and in CB1+/+ mice fed standard chow, which suggests that intestinal CB_1_Rs are not important for gut permeability under lean conditions.

### 3.4. Expression of Gut Barrier and Inflammatory Genes

We next evaluated expression of mRNA for several inflammatory and gut-barrier related genes to determine specific pathways that may be involved in (i) the increases found in intestinal paracellular permeability in mice fed WD for eight weeks and (ii) the impact of genetic deletion of CB_1_Rs in the intestinal epithelium. We developed a custom NanoString panel of 100 genes that included those associated with inflammatory processes, tight junction formation/maintenance pathways, and endocannabinoid system-related components to probe for changes in their expression in the large-intestinal epithelium across diet or genotype. PCA analysis identified unique clustering across both genotypes and diet in the intCB1−/− mice fed WD that was distinct from all other groups. This clustering of samples from the intCB1−/− mice fed WD suggests an effect of both diet and genotype that contributes to an increased inflammatory profile and decreased expression of gut barrier components (Figure 6C). Several DEGs identified in the NanoString Advanced Analysis were further analyzed in the Database for Annotation, Visualization, and Integrated Discovery (DAVID, v6.8). The results identified biological pathways implicated by these DEGs and calculated the number and how strongly associated those genes were with each pathway. Alterations in “molecular functions” were identified that included changes in receptor activity as well as multiple chemokine and cytokine receptor pathways, which may affect the inflammatory response. Several “cellular components” were also highlighted by the analysis and included those involved in anchoring, forming, and maintaining the cell membrane and cell-cell junction. Moreover, several “biological processes” were identified and included those associated with inflammatory processes and those involved in immune cell adhesion and transmigration into tissue. Together, dysregulation of these pathways in mice fed WD for eight weeks, and those with genetic deletion of CB_1_Rs, may contribute to associated changes observed in tight junction formation, inflammatory cell transmigration, and immune cell signaling, which are all key components in gut barrier function [1]. Importantly, intCB1−/− mice maintained on SD displayed largely no changes in gut barrier function or expression of genes associated with gut barrier integrity and local inflammatory responses, which suggests the necessity for a metabolic challenge (i.e., diet-induced obesity) to reveal protective roles for CB_1_Rs in these processes.

Diet-induced gut barrier disruption is associated with changes in expression of genes that control both tight junction integrity (e.g., ZO-1 and occludin) and the inflammatory response (e.g., IL-6, TNFα, and number of T-cells in the large intestine) [4,26]. When compared to mice maintained in SD in the current study, intCB_1_+/+ mice fed WD for eight weeks displayed no changes in expression of genes associated with inflammation in the large intestinal epithelium; however, expression of three claudin-family RNA transcripts were decreased (Figure 6E). In contrast, when comparing intCB1−/− mice fed SD to intCB1−/− mice fed WD, several more DEGs were identified that included several inflammatory-associated genes (Figure 6F). Upregulation of many of these genes is associated with increases in inflammatory cell numbers in tissue (i.e., ICAM1, HLA-DQA1), with an increased state of activation and signaling (i.e., CXCR5/CXCL13, CXCR3, IL1A). Lastly, a similar number of DEGs associated with gut barrier function were found when samples were compared across genotype in mice fed WD. This result suggests that intCB1−/− mice had an increased inflammatory profile with evidence of increased immune cell infiltrate (i.e., HLA-DRA, HLA-DQA1, ICAM1, CD247), and a decrease in expression of genes associated with tight junctions (i.e., TJP2, TJP3, CLDN7). Thus, reduced integrity of the large intestinal barrier may have permitted more pathogen-associated molecules to pass between epithelial cells, thereby stimulating immune cell transmigration and activation in the large intestine. It should be noted that changes in expression of genes do not always translate into changes in expression of protein or function of the associated system. Thus, future studies should examine changes in levels of inflammatory and gut-barrier related proteins under the current conditions as well as to use the current data to guide use of specific genetic mutant models with conditional knockdown of key related proteins to identify roles for the endocannabinoid system in their function.

Differences in DEGs found in the current study when compared to other studies discussed above in rodents maintained on a strictly high-fat diet [4,26] may be due to the composition of the WD used in our study, which contained high levels of sucrose in addition to fats. Indeed, diet composition is a key contributor to microbiota populations that thrive in the gut, and can directly contribute to the inflammatory state of the gut [32,33,34]. Moreover, several studies describe interactions between the endocannabinoid system and the microbiome in the gut [17,18,19,30,35,36,37,38]. Thus, it is plausible that intCB1−/− mice have a unique bacterial composition in the gut that may, in turn, affect gut barrier function and local inflammation. These possibilities will be explored in future studies.

## 4. Materials and Methods

### 4.1. Mice and Diets

Male C57BL/6N (Taconic, Oxnard, CA, USA), conditional intestinal epithelium-specific CB_1_R-deficient (intCB_1_R−/−) [39], and control mice with functional CB_1_Rs in the intestinal epithelium (intCB_1_+/+) mice were given ad libitum access to water and either a standard laboratory rodent diet (SD; Teklad 2020x, Envigo, Huntingdon, UK; 16% kcal from fat, 24% kcal from protein, 60% kcal from carbohydrate as primarily starch) or a Western diet (WD; Research Diets D12079B, New Brunswick, NJ, USA; 40% kcal from fat, 17% kcal from protein, 43% kcal from carbohydrates as mostly sucrose). Conditional intestinal epithelium-specific CB_1_R-null mice [intCB_1_R−/− mice; Cnr1^tm1.1 mrl^/vil-cre ERT2) were generated as previously described by crossing Cnr1^tm1.1 mrl^ mice (Taconic, Oxnard, CA, USA; Model #7599) with vil-cre ERT2 mice (donated by Randy Seeley at University of Michigan, Ann Harbor, MI, USA, with permission by Sylvie Robin at Curie Institute, Paris, France), and have a significant knockdown of Cnr1 gene expression throughout the intestinal epithelium, including in the large-intestinal epithelium (see our [39] for details). Similarly, intCB_1_R+/+ control mice were generated as previously described, and only contained the floxed Cnr1 transgene (i.e., Cnr1^tm1.1 mrl^ [39]). All protocols for animal use and euthanasia were approved by the University of California Riverside Institutional Animal Care and Use Committee (protocol A-20200022) and were in accordance with National Institutes of Health guidelines, the Animal Welfare Act, and Public Health Service Policy on Humane Care and Use of Laboratory Animals. 

### 4.2. Chemicals and Compounds

Dinonadecadienoin (19:2 DAG, Nu-Chek Prep, Waterville, MN, USA) was used as substrate for the DGL assay, and nonadecadienoin (19:2 MAG; Nu-Chek Prep) for the MGL assay. The following compounds were used as internal standards for both lipid extracts and enzyme assays: [^2^H_5_] 2-AG (Cayman Chemical, Ann Arbor, MI, USA) for lipid extracts and the DGL assay, heptadecanoic acid (17:1 FFA; Nu-Chek Prep) for the MGL activity assays, [^2^H_4_]-OEA (Cayman Chemical, Ann Arbor, MI, USA) and [^2^H_4_]-AEA (Cayman Chemical, Ann Arbor, MI, USA) for lipid extracts. JZL 184 (Cayman Chemical, Ann Arbor, MI, USA) was used for MGL inhibition.

### 4.3. Chemical Preparation and Administration

Both intCB_1_R−/− mice and intCB_1_R+/+ mice received intraperitoneal (IP) injections of tamoxifen (Sigma-Aldrich, St. Louis, MO, USA) for five consecutive days as previously described [39]. Fluorescein isothiocyanate (FITC) conjugated to dextran (four kDa; Sigma-Aldrich, St. Louis, MO, USA) was administered via oral gavage in the in vivo gut barrier permeability experiments as detailed below.

### 4.4. In Vivo Gut-Barrier Permeability Assay and Tissue Harvest

To prevent coprophagia, animals were maintained on elevated wire-bottom cages for a 72-h acclimation period and throughout the FITC-Dextran experiment. Food was removed from the cages 4 h prior to gavage with FITC-Dextran and was withheld throughout the experiment. Water was removed after oral gavage with FITC-Dextran to control for metabolism and rate of urination and was returned to the cages immediately after serum was collected (four hours later). At the time of gavage, mice received 0.6 mg/g of 4 kDa FITC-Dextran (100 mg/mL in pure H_2_O). Evidence indicates that nearly all FITC-Dextran would have diffused into the large intestine four hours following oral gavage [40]; therefore, retro-orbital bleeds were performed four hours following administration to collect serum for quantitation of circulating FITC-Dextran as a marker of large intestinal paracellular permeability. At the time of harvest, the epididymal fat pad was removed, rinsed in chilled 1x PBS, dried, and weighed prior to being snap frozen in liquid N_2_. Large intestines were removed, rinsed in chilled PBS, and fecal matter was removed before recording weight and length of the tissue. The large intestine was opened longitudinally and gently washed. Glass slides were used to scrape the intestinal epithelium layer, which was then placed on dry ice, snap frozen in liquid N_2_, and stored at −80 °C until further analysis was performed.

### 4.5. Tissue Lipid Extraction

As previously described [28,41,42,43], frozen large intestine mucosa scrapes were weighed and placed into 1.0 mL of chilled MeOH containing the internal standards [^2^H_5_]-2-AG (500 pmol) [^2^H_4_]-AEA (1 pmol), and [^2^H_4_]-OEA (10 pmol). Lipids were extracted using chloroform (2.0 mL) prior to being washed with 1.0 mL 0.2-micron ultra-purified water. Following centrifugation (1500× *g*, 15 min, 4 °C), the lower organic phase was collected and dried under N_2_ steam (99.998% pure). A second chloroform wash (1.0 mL) was then performed followed by another centrifugation (1500× *g*, 15 min, 4 °C) and collection of the lower phase. Samples were reconstituted in 2.0 mL of chloroform and purified via open-bed silica gel column chromatography. Columns were washed with a 9:1 chloroform:methanol mixture to elute MAGs and FAEs for collection. Collected eluates were dried under N_2_ steam (99.998% pure) and resuspended in 0.2 mL of methanol:chloroform (1:1) prior to analysis via UPLC-MS/MS.

### 4.6. UPLC-MS/MS Analysis of FAEs, MAGs, and Enzyme Assay Products

#### 4.6.1. Quantitation of FAEs and MAGs

Data were acquired using an Acquity I-Class UPLC with direct line connection to a Xevo TQ-S Micro Mass Spectrometer (Waters Corporation, Milford, MA, USA) with electrospray ionization (ESI) sample delivery. Lipids were separated using an Acquity UPLC BEH C_18_ column (2.1 × 50 mm i.d., 1.7 mm, Waters) and inline Acquity guard column (UPLC BEH C_18_ VanGuard PreColumn; 2.1 × 5 mm i.d.; 1.7 mm, Waters), and eluted by an analyte-specific gradient of water and methanol (both containing 0.25% acetic acid, 5mM ammonium acetate). Samples were kept at 10 °C in the sample manager and the column was maintained at 40 °C. Argon (99.998%) was used as the collision gas. Analytes were eluted at a flow rate of 0.4 mL/min and gradient: 80% methanol 0.0–0.5 min, 80–100% methanol 0.5–2.5 min, 100% methanol 2.5–3.0 min, 100–80% methanol 3.0–3.1 min, and 80% methanol 3.1–4.5 min. MS/MS detection was in positive ion mode and capillary voltage set at 0.1 kV. Extracted ion chromatograms were used to quantify AEA (*m*/*z* = 348.3 > 62.0), [^2^H_4_]-AEA (*m*/*z* = 352.3 > 66.1), OEA (*m*/*z* = 326.4 > 62.1), [^2^H_4_]-OEA (*m*/*z* = 330.4 > 66.0), DHEA (*m*/*z* = 372.3 > 91.0), 2-AG (*m*/*z* = 379.3 > 287.3), [^2^H_5_]-2-AG (*m*/*z* = 384.3 > 93.4), 2-DG (*m*/*z* = 403.3 > 11.1), 2-OG (*m*/*z* = 357.4 > 265.2), and 2-LG (*m*/*z* = 355.3 > 263.3). Quantitation occurred using a stable isotope dilution method to detect protonated adducts of the ions [M + H] + in multiple reactions monitoring (MRM) mode. Acyl migration is known to occur in many MAG species following silica-gel purification, therefore the sum of 1-AG and 2-AG, 1-OG and 2-OG, and 1-DG and 2-DG are reported [44].

#### 4.6.2. Quantitation of DGL Assay Product

Data were acquired using the equipment and the elution protocol described above (“Quantitation of FAEs and MAGs”). MS/MS detection was in positive ion mode with capillary voltage maintained at 1.10 kV. Cone voltages and collision energies for respective analytes: 19:2 MAG = 18 v, 10 v; [^2^H_5_]-2-AG = 25 v, 44 v. Lipids were quantified using a stable isotope serial dilution method detecting H^+^ or Na^+^ adducts of the molecular ions [M + H/Na] + in multiple reactions monitoring (MRM) mode (variable amounts of product dinonadecadienoin (19:2 DAG, Nu-Chek Prep, Waterville, MN, USA) versus fixed amount of internal standard [^2^H_5_]-2-AG). Acyl migration from sn-2 to sn-1 positions in monoacylglycerols is known to occur [44,45]; thus, the sum of these isoforms ([^2^H_5_]-1-AG and [^2^H_5_]-2-AG) is presented. Extracted ion chromatograms for MRM transitions were used to quantify analytes: 19:2 MAG (*m*/*z* = 386.4 > 277.2) product of DGL assay and [^2^H_5_]-2-AG (*m*/*z* = 384.3 > 93.4) as internal standard.

#### 4.6.3. Quantitation of MGL Activity Assay Product

Data were acquired using equipment described above (“Quantitation of FAEs and MAGs”). Samples were eluted by a gradient of water and methanol (containing 0.25% acetic acid, 5 mM ammonium acetate) at a flow rate of 0.4 mL/min and gradient: 90% methanol 0.0–0.1 min, 90–100% methanol 0.1–2.0 min, 100% methanol 2.0–2.1 min, 100–90% methanol 2.1–2.2 min, and 90% methanol 2.2–2.5 min. MS detection was in negative ion mode with capillary voltage maintained at 3.00 kV. Cone voltages for nonadecadienoic acid (19:2 FFA; Nu-Chek Prep, Waterville, MN, USA) = 48 v and heptadecanoic acid (17:1 FFA; Nu-Chek Prep, Waterville, MN, USA) = 64 v. Lipids were quantified using a dilution series detecting deprotonated molecular ions in selected ion reading (SIR) mode (variable amounts of product 19:2 FFA versus fixed amount of internal standard 17:1 FFA). Extracted ion chromatograms for SIR masses were used to quantify analytes: 19:2 FFA (*m*/*z* = 293.2) product of MGL enzyme assay and 17:1 FFA (*m*/*z* = 267.2) as internal standard.

### 4.7. DGL and MGL Activity Assays

#### 4.7.1. Protein Isolation

As previously described [28,41,42,43], samples homogenized in 2 mL of chilled 50 mM Tris–HCl, 320 mM sucrose buffer (pH = 7.5 at 37 °C) were centrifuged at 800× *g* for 15 min at 4 °C. The supernatant was collected in a 2.0 mL centrifuge tube and sonicated twice for 10 s prior to two sequential freeze thaw cycles. Centrifuged (800× *g*, 15 min, 4 °C) supernatants were then quantified via BCA assay and normalized between all samples.

#### 4.7.2. DGL Activity Assay

Normalized large intestine protein samples (100 mL) were incubated at room temperature for 10 min with the MGL inhibitor JZL 184 to inhibit metabolism of the product of interest. These samples were then incubated at 37 °C for 30 min with 20 nmol/reaction of the DGL substrate 19:2 DAG in buffer (50 mM Tris-HCl, 0.2% Triton x-100, pH = 7.0 at 37 °C). The reaction was stopped by the addition of 1.0 mL of chilled methanol containing 25 pmol of the internal standard [^2^H_5_]-2-AG. Lipids were extracted as described above (“Tissue Lipid Extraction”) and quantified as described above (“UPLC-MS/MS Analysis of FAEs, MAGs, and Enzyme Assay Products”).

#### 4.7.3. MGL Activity Assay

Normalized large intestine protein samples (100 mL) were incubated at 37 °C for 10 min with 50 nmol/reaction of the MGL substrate19:2 MAG in buffer (50 mM Tris-HCl, 0.25% Fatty-Acid Free BSA, pH = 8.0). The reaction was stopped by the addition of 1.0 mL of chilled methanol containing 5 nmol/reaction of the internal standard 17:1 FFA. Lipids were extracted as described above (“Tissue Lipid Extraction”) and quantified as described above (“UPLC-MS/MS Analysis of FAEs, MAGs, and Enzyme Assay Products”).

### 4.8. Large Intestine Imaging

At the time of harvest, the large intestine was removed and flushed with 4% Paraformaldehyde (PFA) and then stored in a 4% PFA solution over-night. The large intestine was then transferred to a 30% sucrose solution for 48 h, blocked in OCT, and sliced at 5 µm thick sections for staining. Hematoxylin and eosin staining and imaging under a DM5500B microscope (Leica) were performed as previously described [46].

### 4.9. RNA Analysis

#### 4.9.1. RNA Isolation and Quantitative Real-Time PCR (qPCR)

Total RNA was isolated from large intestine mucosal scrapes using a RNeasy kit (Qiagen, Valencia, CA, USA) and first-strand cDNA was generated using M-MLV reverse transcriptase (Invitrogen, Carlsbad, CA, USA). PrimePCR Assays (Biorad, Irvine, CA, USA) were used to perform quantitative RT-PCR including primers for CB_1_R (Cnr1), Tight junction protein-1 (Tjp-1), Occludin (Ocln), and Claudin-1 (Cldn-1) gene transcripts under preconfigured SYBR Green assays (Biorad, Irvine, CA, USA). Identification of differential expression between intCB_1_R−/− and intCB_1_R+/+ control mice was performed using the common delta-delta (2^−ΔΔCq^) method with Hprt as the housekeeping gene [47].

#### 4.9.2. NanoString nCounter Gene Expression Assays and Analysis

nCounter gene expression assays (NanoString Technologies, Seattle, WA, USA) were performed using a custom NanoString Panel which probed for 100 genes. Custom panel codeset probes were hybridized with 500 ng of total RNA per large intestine mucosa sample for 18 h at 65 °C according to NanoString protocol. Molecular grade water was used to dilute hybridized RNA which was then loaded into nCounter SPRINT cartridge (NanoString, Seattle, WA, USA), and quantified. NanoString Sprint Profiler technology was utilized to count RNA-conjugated probes.

#### 4.9.3. nSolver and Advanced Analysis

Results from the panel were normalized in nSolver following best practices and analyzed using nSolver (v4.0, NanoString, Seattle, WA, USA) and Advanced Analysis (NanoString, Seaalte, WA, USA) software according to previously published protocols [48,49]. Principal Component Analysis (PCA) plots were generated using the nSolver software and raw barcode counts. nSolver-generated heat maps were created using normalized data and agglomerative clustering, a bottom-up form of hierarchical clustering (NanoString User Manual C0019-08, Seattle, WA, USA). For Advanced Analysis, normalized data were used (NanoString User Manual 10030-03, Seattle, WA, USA). Differential expression (DE) analysis was performed to identify specific targets that exhibit significantly increased or decreased expression compared to control group indicated in the figure. Summary pathway score plot colors are based on calculated scores and are represented as down regulation (yellow) to upregulation (blue).

### 4.10. Statistical Analysis

Data were analyzed using GraphPad Prism7 (Dotmatics, Boston, MA, USA) software using unpaired Student’s *t*-tests (two-tailed) and regular or repeated measures two-way or three-way analysis of variances with Holm-Sidak’s multiple comparisons post-hoc test when appropriate. Differentially expressed genes (DEGs) were defined by a *p*-value < 0.05 when an unpaired one-tailed *t*-test with Welch’s correction was performed (RRID:SCR_002798). Gene enrichment and functional annotation analyses of DEGs were performed using open-source functional enrichment tool: Database for Annotation, Visualization, and Integrated Discovery (DAVID, v6.8, Laboratory of Human Retrovirology and Immunoformatics, Washington, DC, USA, accessed: 17 May 2021) [50,51]. All data sets expressed as mean ± standard error (SEM) and significance was determined as *p* < 0.05.

## 5. Conclusions

Together, these data provide evidence of an important role for CB_1_Rs in the intestinal epithelium in diet-induced dysregulation of gut permeability and associated inflammation. Future experiments should consider mechanisms by which overactivation of intestinal epithelial CB_1_Rs influences epithelial cell-cell junctions, and to identify if the endocannabinoid system participates in function of the master regulators of the tight junction complex. Moreover, it will be important to identify specific time points between week two and eight of exposure to WD that gut barrier permeability becomes compromised, and to determine if this precedes the enhanced inflammatory profile found at week eight.

## Figures and Tables

**Figure 1 ijms-23-10549-f001:**
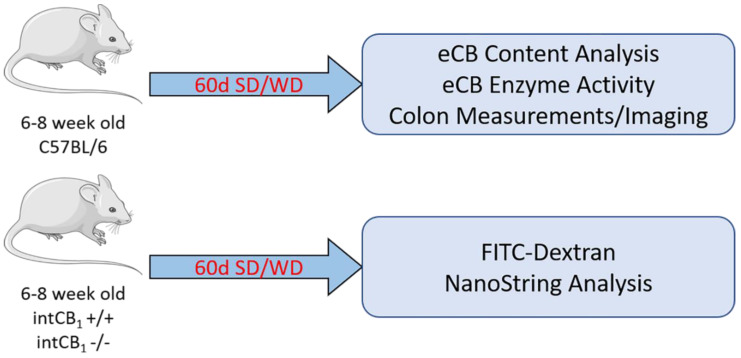
Experimental design. Male wild-type (WT), intCB1−/−, and intCB1+/+ mice were placed on a SD or a WD for eight weeks. Levels of endocannabinoids (eCBs) and activity of their metabolic enzymes were analyzed in WT mice (top panel), and gut-barrier function and analysis of expression of genes associated with gut barrier function and inflammation was performed on SD- and WD-fed intCB1+/+ control mice and intCB1−/− mice (bottom panel). SD = standard diet, WD = Western diet, intCB1−/− = conditional intestinal epithelial cannabinoid receptor subtype-1 deficient mice.

**Figure 2 ijms-23-10549-f002:**
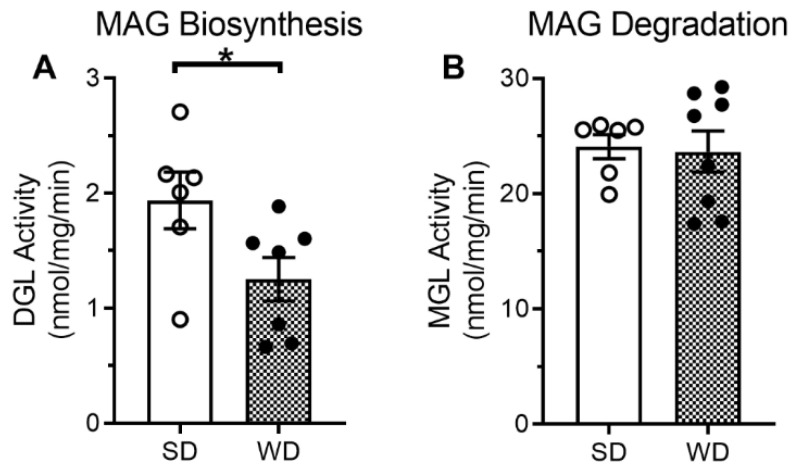
Changes in endocannabinoid enzymatic activity in the large-intestinal epithelium in mice fed WD. Mice fed WD for eight weeks had a significant reduction in the activity of MAG biosynthetic enzymes in the large-intestinal epithelium (**A**) with no changes in activity of the MAG degradative enzymes (**B**) when compared to mice fed SD. Data analyzed by two-tailed unpaired Student’s *t*-test; * = *p* < 0.05, *n* = 6–8 per condition. SD = standard diet, WD = Western diet.

**Figure 3 ijms-23-10549-f003:**
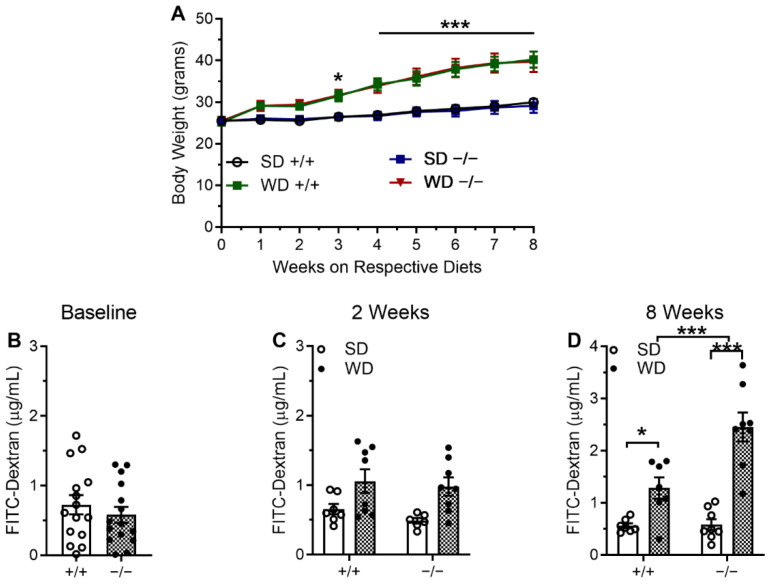
Large-intestinal permeability is increased in mice fed WD and further exacerbated in intCB1R−/− mice fed WD. Independent of genotype, mice fed WD gained significantly more weight when compared to mice fed SD (**A**). intCB1R−/− mice displayed no changes in baseline paracellular permeability when compared to intCB1R+/+ control mice (**B**). Exposure to WD for two weeks did not induce significant changes in paracellular permeability in intCB1R+/+ control mice or intCB1R−/− mice (**C**). Exposure to WD for 8 weeks in intCB1R+/+ control mice, led to increases in large-intestinal paracellular permeability that was exacerbated in intCB1R−/− mice (**D**). Baseline FITC-Dextran data analyzed via unpaired two-tailed Student’s *t*-test. Body weight data analyzed via three-way ordinary ANOVA with effects of diet F(1, 252) = 211.3, *p* < 0.0001; time F(8, 252) = 24.42, *p* < 0.0001; and no effect of genotype F(1, 252) = 0.003378, *p* = 0.9537; Tukey’s multiple comparisons test performed as post hoc analysis. FITC-Dextran data analyzed via two-way ANOVA at 2 weeks with an effect of diet F(1, 25) = 12.43, *p* = 0.0017; no effect of genotype F(1, 25) = 0.9498, *p* = 0.3391; and no effect of interaction F(1, 25) = 0.1312, *p* = 0.7202. FITC-Dextran data analyzed via two-way ANOVA at 8 weeks with an effect of diet F(1, 26) = 48.39, *p* < 0.0001; genotype F(1, 26) = 10.30, *p* = 0.0035; and interaction F(1, 26) = 9.485, *p* = 0.0048; Holm-Sidak performed as post hoc analysis. * = *p* < 0.05, *** *p* < 0.001; *n* = 4–9 (**A**), *n* = 15 (**B**), *n* = 6–8 (**C**,**D**). per condition. SD = standard diet, WD = Western diet, +/+ = intCB1+/+ mice, −/− = intCB1−/− mice.

**Figure 4 ijms-23-10549-f004:**
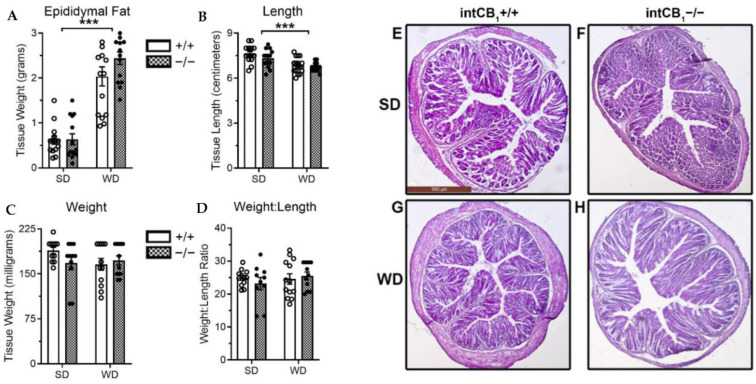
Effects of genotype and diet on epididymal fat mass, and large intestine length and weight. Mice fed WD for eight weeks displayed significant increases in epididymal fat mass (**A**) and significant decreases in colon length (**B**) regardless of genotype. However, no changes in colon weight or weight:length ratio were observed across diet or genotype (**C**,**D**). No significant alterations were observed in tissue architecture in representative 4× images of 5 µm thick colon slices (**E**–**H**). Epididymal fat data analyzed via two-way ordinary ANOVA with an effect of diet F(1, 53) = 114.8, *p* < 0.0001; no effect of genotype F(1, 53) = 1.998, *p* = 0.1633; and no effect of interaction F(1, 53) = 1.640, *p* = 0.2059; Holm Sidak multiple comparisons test performed as post hoc analysis. Large intestine length data analyzed via two-way ordinary ANOVA with an effect of diet F(1, 53) = 23.61, *p* < 0.0001; no effect of genotype F(1, 53) = 2.358, *p* = 0.1306; and no effect of interaction F(1, 53) = 0.4664, *p* = 0.4976; Holm Sidak multiple comparisons test performed as post-hoc analysis. Large intestine weight data analyzed via two-way ordinary ANOVA with no effect of diet F(1, 42) = 1.169, *p* = 0.2857; no effect of genotype F(1, 42) = 0.6613, 9 = 0.4207; and no effect of interaction F(1, 42) = 2.479, *p* = 0.1229. Large intestine weight:length ratio data analyzed via two-way ordinary ANOVA with no effect of diet F(1, 42) = 0.5132, *p* = 0.4777; no effect of genotype F(1, 42) = 0.07802, *p* = 0.7814; and no effect of interaction F(1, 42) = 0.9553, *p* = 0.3340. *** = *p* < 0.001, *n* = 11–15 per condition. SD = standard diet, WD = western diet, +/+ = intCB1+/+ mice, −/− = intCB1−/− mice.

**Figure 5 ijms-23-10549-f005:**
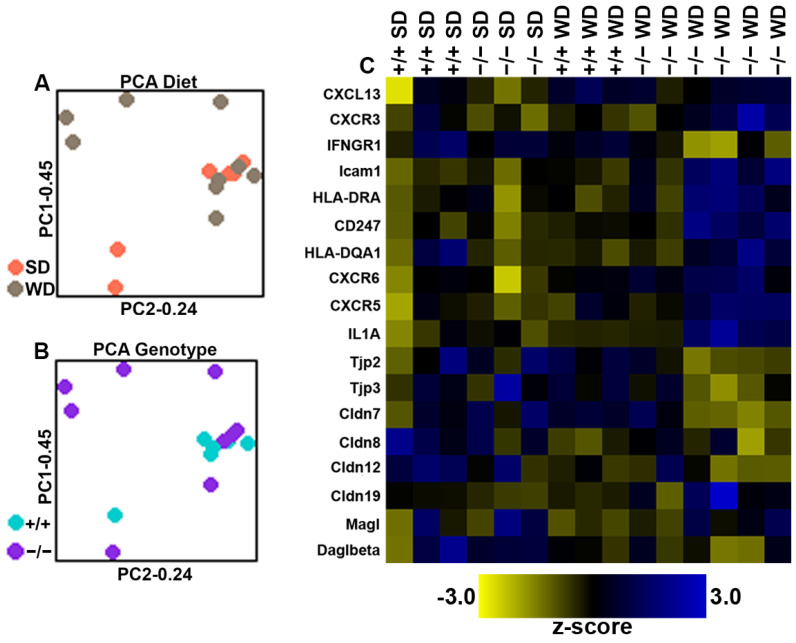
Principal Component Analysis (PCA) of expression of genes in the large-intestinal epithelium. PCA was performed across both diet (**A**) and genotype (**B**) to identify unique group clustering within the 100 probed genes in the custom NanoString panel. A representative heatmap of all differentially expressed genes (DEGs) in the large intestine mucosa across either diet or genotype revealed increases in inflammatory transcripts and a downregulation in barrier-associated transcripts in intCB1R−/− mice fed WD (**C**). SD = standard diet, WD = western diet, +/+ = intCB1+/+ mice, −/− = intCB1−/− mice.

**Figure 6 ijms-23-10549-f006:**
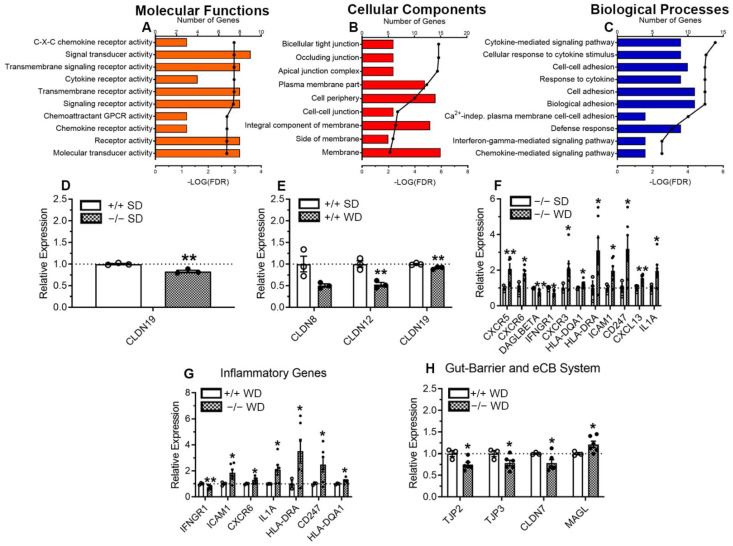
Pathways affected by genotype and diet. DEGs were entered into the Database for Annotation, Visualization, and Integrated Discovery (DAVID, v6.8), which revealed several pathways that may be implicated in diet-induced gut barrier disruptions in cytokine and chemokine signaling pathways in molecular functions (**A**), cell-cell junctions and barrier functions in cellular components (**B**), and cytokine and cellular adhesion functions in biological processes (**C**). intCB1−/− mice fed SD displayed a down regulation only in Cldn19 when compared to intCB1+/+ mice fed SD (**D**). intCB1+/+ mice fed WD displayed a downregulation of genes for three claudins when compared to intCB1+/+ mice fed SD (**E**) and intCB1−/− mice fed WD displayed dysregulation of 11 genes associated with inflammatory and eCB system function (**F**). Similarly, 11 DEGs were observed in intCB1−/− mice fed WD when compared to intCB1+/+ mice fed WD including an upregulation of several inflammatory genes (**G**) and a down regulation of a several genes related to gut barrier function; however, expression of genes for MGL was increased (**H**). Data analyzed via unpaired one-tailed t-test with Welch’s correction. * = *p* < 0.05, ** = *p* < 0.01, *n* = 3–6 per condition. SD = standard diet, WD = Western diet, +/+ = intCB1+/+ mice, −/− = intCB1−/− mice.

**Table 1 ijms-23-10549-t001:** UPLC/MS/MS analysis of endocannabinoid content in the large intestinal epithelium of mice fed SD or WD. MAGs (2-AG, 2-DG, 2-LG, 2-OG) represented as nmol per g tissue; FAEs (AEA, DHEA, OEA) represented as pmol per g tissue. Data analyzed via two-tailed unpaired student’s *t*-test; * = *p* < 0.05, ** = *p* < 0.01; *n* = 5–8 per condition. SD = standard diet, WD = western diet.

Analytes	SD	WD	*p*-Value
2-AG	24.08 ± 2.03	17.99 ± 1.58	0.033 *
2-DG	5.65 ± 1.62	1.83 ± 0.47	0.018 *
2-LG	181.79 ± 42.38	40.73 ± 11.36	0.003 **
2-OG	97.68 ± 17.21	74.17 ± 17.34	0.366
AEA	10.04 ± 1.89	10.14 ± 1.88	0.971
DHEA	34.76 ± 3.68	35.65 ± 2.8	0.848
OEA	343.4 ± 57.93	156.7 ± 9.33	0.005 **

## Data Availability

The data presented is contained within the article or Appendix A.

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
