# Peer review of "Diet-Induced Gut Barrier Dysfunction Is Exacerbated in Mice Lacking Cannabinoid 1 Receptors in the Intestinal Epithelium"

_ijms, 2022, doi:10.3390/ijms231810549_

Round 1
Reviewer 1 Report
In the current manuscript the authors are investigating the role of the cannabinoid receptor subtype 1 (CB1R) in western diet induced gut dysfunction. The authors previously generated intCB1-/- mice, which have a conditional deletion of CB1R in the intestinal epithelium. (It is unclear why these mice are not called intCB1R-/- or CNR1-/-). In this study they compared the effect of a western diet (WD) on gut barrier, levels of endocannabinoids and expression of junction proteins and inflammatory markers in the large intestine of intCB1-/- and CB1+/+ mice. They demonstrate that on a WD: (1) WT mice have lower levels of endocannabinoids and (2) intCB1-/- mice have increased gut permeability compared with CB1+/+ mice. These changes are in the large intestine.
The endocannabinoid system plays an important role in regulating food intake and gut physiology. However, our current understanding is limited and controversial. Thus, it is important to have good model systems and carefully conducted research to determine the function of the endocannabinoid system. The authors have previously generated intCB1-/- mice, which is an important step. However, instead of conducting a detailed study on these mice they published some data in an earlier manuscript and the current manuscript is a follow up. Unfortunately, in the current study the only major change they identify is an increase in gut permeability in intCB1-/- mice on a WD. The authors have previously looked at similar parameters in the small intestine, whereas here they focus on the large intestine. They don’t provide a justification for this. Some of the data in the manuscript are a re-presentation of previous published data. For the RNA data the changes in expression for many of the genes is not biologically significant.
Results: The numerical data is included in the text, which is unusual, disrupts the text and makes the reading cumbersome. It is also included in the Supplementary tables. Please remove all numerical data from the text.
Fig.2: an increase in body weight on a WD compared with a standard diet is not novel. The authors have themselves published similar data in their Physiol Behav. 2017 paper and has been shown by other investigators.
Fig. 2B and C: Why was the DGL and MGL activities not determined in intCB1-/- mice?
Table 2: Why were levels of endocannabinoids not determined in intCB1-/- mice? Changes in levels of encannabinoids due to a WD is not novel.
The P values should be included in this table instead of having a separate supplementary table for the same data.
Fig. 3A: shows no difference in weight gain between intCB1-/- and CB1+/+ mice on SD or WD. Once again, this data is not truly novel. In their Nutrients, 2020 paper the authors show that there is no difference in total caloric intake between +/+ and -/- mice on WD (though they don’t actually show weight gain).
Figs. 5 and 6 show changes in gene expression for 18 genes (not 19 as mentioned in the text), which were identified by NanoString Sprint Profiler technology. The heat map in Fig 5 and the graphs in Fig. 6 are showing the same data in different ways. Please do not show the same data in more than one way. Many of the genes shown in these figures and described as being differentially expression are likely not biologically relevant. A general rule for changes in gene expression is that the change should be ³2-fold to be biologically significant.
Supplementary Table 4 shows the numerical data for the heat map. The sequence of genes is different between the heat map and the table. Please keep the same sequence. The numerical data is also repeated in the Results section, which is unnecessary and makes reading cumbersome.
Author Response
Our responses can be found in the attached word document. Our responses are italicized.

Reviewer 2 Report
This manuscript observes Western diet-induced obesity mouse mode and transgenic mice conditionally deficient in CB1Rs (intCB1-/-) in the intestinal epithelium to determine the effect of the diet-induced endocannabinoid system in the intestinal epithelium in intestinal barrier dysfunction. The overall experimental purpose is clear, the methodology is appropriate, the data quality is good, and the results are adequately explained. The discussion also clearly compares the studies with different conclusions related to this field, adding to the important contribution of this manuscript. This manuscript information is ideal for publication.
Author Response
Thank you for these very positive comments!